# Chlorhexidine Mucoadhesive Buccal Tablets: The Impact of Formulation Design on Drug Delivery and Release Kinetics Using Conventional and Novel Dissolution Methods

**DOI:** 10.3390/ph14060493

**Published:** 2021-05-23

**Authors:** Enas Al-Ani, David Hill, Khalid Doudin

**Affiliations:** 1Research Institute in Healthcare Science, Faculty of Science and Engineering, University of Wolverhampton, Wolverhampton WV1 1LY, UK; 2School of Biology, Chemistry and Forensic Science, Faculty of Science and Engineering, University of Wolverhampton, Wolverhampton WV1 1LY, UK; 3Department of Chemistry, The University of Sheffield, Sheffield S10 2TN, UK; K.Doudin@sheffield.ac.uk

**Keywords:** chlorhexidine, mucoadhesive, hydrogel, buccal, release kinetics, flow rate, *Candida albicans*, cytocompatibility

## Abstract

Oropharyngeal candidiasis (OPC) is a mucosal infection caused by *Candida* spp., and it is common among the immunocompromised. This condition is mainly treated using oral antifungals. Chlorhexidine (CHD) is a fungicidal and is available as a mouth wash and oral gel. It is used as an adjuvant in the treatment of OPC due to the low residence time of the current formulations. In this study, its activity was tested against *C. albicans* biofilm and biocompatibility with the HEK293 human cell line. Then, it was formulated as mucoadhesive hydrogel buccal tablets to extend its activity. Different ratios of hydroxypropyl methylcellulose (HPMC), poloxamer 407 (P407), and three different types of polyols were used to prepare the tablets, which were then investigated for their physicochemical properties, ex vivo mucoadhesion, drug release profiles, and the kinetics of drug release. The release was performed using Apparatus I and a controlled flow rate (CFR) method. The results show that CHD is biocompatible and effective against *Candida* biofilm at a concentration of 20 µg/mL. No drug excipient interaction was observed through differential scanning calorimetry (DSC) and Fourier-transform infrared spectroscopy (FTIR). The increase in P407 and polyol ratios showed a decrease in the swelling index and an increase in CHD in vitro release. The release of CHD from the selected formulations was 86–92%. The results suggest that chlorhexidine tablets are a possible candidate for the treatment of oropharyngeal candidiasis.

## 1. Introduction

Oropharyngeal candidiasis (OPC) is a common opportunistic infection in immunocompromised patients, caused by *Candida* spp. and, most commonly, *C. albicans*. The annual estimated number of OPC cases is 10 million globally. The treatment of OPC is hampered by the limited number of antifungal drugs available, drug resistance, drug–drug interaction, and adverse effects. Uncontrolled OPC could result in systemic candidiasis, which has a mortality rate of 30–50% [1,2].

Systemic antifungals are the most effective treatment for OPC, specifically azoles (fluconazole, itraconazole, miconazole, clotrimazole) and polyene (nystatin or amphotericin B deoxycholate) [3]. Antifungals cause a variety of side effects, including gastrointestinal disturbances, nephrotoxicity, and hepatotoxicity [4]. Furthermore, azoles exhibit drug–drug interaction by inhibiting the CYP3A4 isoenzyme, thus, increasing the risk of immunosuppressant toxicity due to their low therapeutic index of the latter [5]. Another drawback of treatment with azoles is that *Candida* can develop resistance due to their fungistatic activity rather than fungicidal [6]. Caspofungin is recommended as a second line treatment for patients refractory to azoles and polyenes [7]. It is administered as an intravenous infusion due to its poor intestinal absorption and short half-life, thus, limiting its administration to hospitalized patients only [8]. Furthermore, immunocompromised patients with OPC developed systemic candidiasis after they had received nystatin, which failed to control the infection [9]

Cellulose derivatives, chitosan, and polyacrylic acid derivatives are the main hydrogel-forming polymers used to prepare mucoadhesive buccal dosage forms due to their mucoadhesive properties, biocompatibility, cost-effectiveness, and availability [10,11,12]. Several hydrogel-forming polymers have been previously investigated for the treatment of OPC during the past decades. For instance, chitosan, pectin, and HPMC were formulated as miconazole buccal films for the delivery of miconazole nitrate [13]. Fluconazole oral strips were prepared with HPMC and polyacrylic acid derivatives [10]. Both formulations showed more than 80% drug release in the first hour. A CHD buccal tablet prepared with HPMC, carbopol, and lactose sustained the release for up to eight hours [14]. Chitosan and sodium alginate buccal film successfully extended the release of CHD up to three hours [15]. Clotrimazole and nystatin were loaded individually into hydroxypropyl cellulose and polyethylene oxide films. The release of the drug was maintained for up to eight hours [16].

To combat drug resistance and overcome systemic side effects, chlorhexidine (CHD) was selected in this study to be formulated as a local controlled drug delivery. CHD is available as a 0.2% *w/v* mouthwash, a 0.2% *w/v* oral spray, and a 1% *w/v* dental gel. However, they all have a short retention time in the oral cavity. As CHD cannot be absorbed from the gastrointestinal tract, it has no systemic side effects [17], and to date, there are no documented reports of acquired resistance to CHD [18]. Antifungal activity is achieved through the binding of CHD cations to the anionic surface charge of the fungal cell wall, leading to a decrease in adhesion capacity, the loss of structural integrity, and finally, disruption of the cell wall [19]. Consequently, there is an urgent need for a new approach to the localized, prolonged delivery of antifungal agents for enhanced OPC therapy.

The aim of this study was to prepare safe and effective chlorhexidine mucoadhesive hydrogel buccal tablets containing HPMC and P407 polymers. The tablets were designed to control the release of CHD over two hours to minimize patient intolerance that can result from a prolonged application time and lead to withdrawal from treatment [20]. Drug release studies were conducted using dissolution Apparatus 1 and a newly-designed dissolution method to investigate the release of CHD using a controlled rate of dissolution media at 1 mL/min to mimic drug release in the oral cavity, which is performed by saliva. The normal salivary flow rate is ≥1 mL/min with a maximum value of 7 mL/min [21,22]. Despite there being several marketed buccal tablets, there is no compendial standard available to mimic drug release in the oral cavity.

## 2. Results and Discussion

### 2.1. Antifungal Activity of CHD

The minimum inhibitory concentration (MIC) and minimum biocidal concentration (MBC) of CHD against *C. albicans* planktonic cells were 2.5 and 5 µg/mL, respectively. The effect of CHD was investigated for two hours against both immature and mature biofilms. The investigation was performed to establish the concentration of CHD against *C. albicans* biofilms to be incorporated in the tablets. The viability of CHD-treated biofilms was measured using an XTT reduction assay. CHD showed nearly a 100% loss of mitochondrial activity on a 4-h biofilm at a concentration of ≥10 µg/mL, and the viability was around 16% at a concentration of 5 µg/mL (Figure 1a). The effectiveness of CHD decreased with an increase in the maturity of the biofilm (Figure 1b). Nearly a 100% suppression of metabolic activity was achieved at a concentration of ≥40 µg/mL with a significant reduction to 15% and 26% at 20 and 10 µg/mL of CHD, respectively.

The sensitivity of planktonic cells to antifungals is higher than that of biofilm cells. This is because there is a more uniform distribution of the drug in planktonic cell culture media, whereas a concentration gradient of extrinsic and intrinsic material results from the uneven thickness and cell density in the biofilm [23]. The increase in the resistance of a pathogenic strain of *C. albicans* biofilm with an increase in its maturity and using CHD has previously been documented. The MIC increased from 16 to 256 µg/mL for ~4-h and 72-h biofilms, respectively. This increase in resistance might be due to extracellular material, genetic or biochemical changes in the cells [24].

### 2.2. Biofilm Recovery 

The recovery of the biofilm was measured using a viable count of 24-h biofilms treated with CHD. It was undertaken to determine the number of living cells that were capable of reproduction. The recorded viability was 4% at a concentration of ≥40µg/mL compared to 0.5% for the XTT assay (Figure 1c). The recovery was 26.8% and 62.8% compared to 15.4% and 26.6% mitochondrial activity at 20 and 10 µg/mL, respectively. This means that XTT overestimates the killing effect at certain concentrations, which might be attributed to the low sensitivity of XTT towards slow-growing *candida* cells that are affected by drug treatment [25].

### 2.3. Cytocompatibility

Neutral red is a weak basic dye that penetrates cells by diffusion and accumulates in the lysosomes, staining them red due to their low pH compared to the cytoplasm. In living healthy cells, the pH gradient is maintained. However, when a cell dies, lysosomes are not able to retain the dye due to the loss of the pH gradient [26]. NR assay was used to measure the lysosomal activity of human cells and, consequently, facilitate the estimation of the cytocompatibility of the drugs. CHD is considered to be cytotoxic against HEK293 cells at concentrations of 80 and 160 µg/mL (Figure 2). The retained lysosomal activity was less than 20%. Drugs are considered cytotoxic when cell viability is ≤70% [27]. At a concentration of ≤20 µg/mL of CHD, the preserved viability was approximately 100%.

Based on the microbiological investigations, a concentration of 20 µg/mL of CHD was effective in killing planktonic cells and 4-h biofilm cells and 26.8% of the mature biofilm survived. This concentration is considered cytocompatible with HEK293 cells; they showed a viability of 96.5%. Consequently, a concentration of 20 µg/mL of CHD was chosen for the formulation of the tablets due to its efficacy and safety. Moreover, upon repeated application, new biofilm formation will be inhibited, and mature biofilms will be diminished and eradicated over time.

### 2.4. Characterisation of Powder Blends and Granules

In the current study, hydrogel-forming polymers were used to formulate a mucoadhesive buccal tablet to prolong and control the release of CHD in the oral cavity. Hydrogels comprise a three-dimensional, crosslinked structure of hydrophilic polymers, which retain water and form porous structures [12]. They are used as drug delivery systems as they resemble natural tissue, in that they retain large quantities of water [28,29]. Three groups of CHD hydrogel mucoadhesive buccal tablets were prepared based on the type of polyol: sorbitol, mannitol, and xylitol. Each group was prepared with two different ratios of polymers (P407 to HPMC) and two different ratios of polyols (Table 1). The ratio of the polymers was chosen on the basis of our previous investigation [30]. Granules were prepared using melt granulation; both the excipients and CHD are thermostable as demonstrated by DSC analysis (see Appendix A)

HPMC is a non-ionic hydrophilic polymer with mucoadhesive and hydrogel-forming properties. It has widely been investigated in buccal drug delivery and is included in marketed products [31]. P407 is a non-ionic co-polymer of polyethylene oxide and polypropylene oxide that has a thermo-reversible, hydrogel-forming ability. It has mucoadhesive properties and gels at a concentration of > 20% at 25 °C [32]. It has wide application in drug delivery, for example, in oral solutions; suspensions; inhalers; and parenteral, ophthalmic, and topical formulations. Its mucoadhesive potential has been utilized in rectal and ophthalmic preparations [33]. In the current study, it was formulated into tablet dosage forms with HPMC to control the release of CHD by taking advantage of its surfactant and hydrogel-forming properties to improve the hydration of the tablet and maintain the three dimensional structure [34].

Polyols have a sweet taste and provoke a cooling sensation resulting from their negative heat of dissolution. They have a low glycemic index and are non-cariogenic [35]. They are frequently used in oral dosage forms, such as orally disintegrating tablets, lozenges, chewable tablets, chewing gum toothpaste [36], and as moisturizers in artificial saliva [37]. Sorbitol, mannitol, and xylitol are sugar alcohols, which do not promote or reduce tooth decay [38]. Sorbitol, mannitol, and xylitol were added to improve the taste perception, especially because they have non-cariogenic properties.

Flow through an orifice is proposed as a better method to measure powder flowability based on the British and United States Pharmacopoeias. The mass flow of the powders and granules was investigated by measuring the flow through a funnel with an orifice size of 10 mm diameter f. Powder blends failed to pass through the orifice. However, all granules successfully passed through the orifice (Table 2), and formulations with a higher ratio of P407 showed an improvement in mass flow. Based on the compressibility index, the flowability of powder blends varied from ‘poor’ to ‘very very poor’. The flowability of granules displayed a remarkable improvement, ranging from ‘passable’ to ‘good’ (Table 2) [39].

#### 2.4.1. Physical Properties of the Tablets

Table 3 presents the friability and tensile strength results. The tablets showed an acceptable level of friability of less than 1%. Within each group of formulations, there was no significant difference in tensile strength (*p* > 0.05). However, the only reported difference between the three groups was between (S1and X1) and (S1and M1) (*p* < 0.05), with S1 having a higher tensile strength. Sorbitol displayed plastic deformation and has good binding properties [40].

#### 2.4.2. Swelling Index (SI)

The swelling index of the tablets was directly proportional to the HPMC ratio and inversely proportional to the P407 and polyol ratios. This is attributed to the higher molecular weight of HPMC compared to P407 and polyols. The decrease in SI with an increase in the polyols ratio is explained by their higher solubility [41]. There is a significant difference in SI (*p* < 0.5) within each group of formulations. However, the SI of S4, M4, and X4 was 3 with no significant difference (*p* > 0.5), which might indicate that the type of sugar has no impact on the swelling of the formulations. The swelling profile for all formulations is presented in the Appendix A).

#### 2.4.3. Determination of Ex Vivo Residence Time

The disintegration apparatus was used to test the residence time of the adhered tablets to the tissue. The frequency of the moving arm of the disintegration apparatus was 30 cycles per minute, which is equivalent to compendial disintegration testing for tablets. All tested tablets successfully adhered to the chicken crop (pouch), which was repeatedly immersed in the aqueous media for two hours. This represents the ideal residence time of the tablets in the buccal cavity. Figure 3 shows the residue of S4, M4, and X4 tablets at the end of the test. The mucoadhesive property is attributed to the presence of HPMC and P407 [42,43].

#### 2.4.4. In Vitro Dissolution and Erosion Studies

Despite there being several marketed buccal tablets, there is no compendial standard for drug release in the oral cavity. Two methods of dissolution were used to investigate the release of CHD: (i) Apparatus 1, using 500 mL of dissolution media: this method was used to test the effect of a large volume of dissolution media on drug release, and to investigate the effect of fluid consumption while the table is adhering to the oral mucosa. (ii) drug release based on a controlled flow rate (CFR) of 1 mL/min to mimic the salivary flow rate in the oral cavity. The normal salivary flow rate is ≥1mL/min with a maximum value of 7 mL/min [21,22]. Drug release based on a controlled flow rate has previously been investigated elsewhere [44,45,46,47].

Dissolution results are presented in Figure 4: Apparatus I (a, b, and c) and CFR (d, e, and f). Using Apparatus I, all formulations showed a release of >60%. However, release using the CFR method for the same formulations was affected by the ratio of polymers. The release of CHD from the tablets with equal amounts of HPMC and P407 (S1,S2,M1,M2,X1, and X2) was around 30%. However, the release was >60% from tablets with a HPMC to P407 ratio of 1:3 (S3,S4,M3,M4,X3, and X4).

The release of CHD using Apparatus 1 was 93% for M4 and X4 and 87% for S4 and, using CFR, the release was 87, 90, and 89% for S4, M4, and X4, respectively. An unpaired T-test showed no significant difference between the two methods (*p* > 0.05). This is attributed to the higher hydrophilicity of these formulations, which resulted in a faster hydration of the matrices and rapid formation of the interlocked gel layer in S4, M4, and X4. The gel layer prevents the rapid ingress of water and subsequently controls drug release from the hydrogel matrix [48]. The increase in the ratio of sorbitol, mannitol, and xylitol improved CHD release from S3 to S4 due to their solubility, which was enhanced by the surfactant activity of P407 [33,49], and there was a similar improvement for mannitol and xylitol correspondent formulations. P407 improved the release of CHD from the HPMC matrix, and this is attributed to the absence of interaction between the polymers, CHD, or polyols in S4, M4, and X4 formulations [42]. This was confirmed by the DSC and FTIR results, which showed no interaction between CHD and the excipients of the formulations (see Appendix A).

The non-cumulative CHD released from S4, M4, and X4 using a constant flow rate are presented in Figure 5. CHD release showed a slight burst of release at a concentration of 22–23 µg/mL at the 10-min point, followed by a concentration range of 16–21 µg/mL for all three formulations. Data relating to non-cumulative release from other formulations are available in the Appendix A. Based on the cytotoxicity and antifungal assays, this range of concentration is considered safe and effective, and upon reapplication of the tablets, CHD can be used for the treatment of OPC. However, further in vivo investigation needs to be conducted.

In order to understand the mechanism of drug release, tablet erosion (E%) was investigated. This was determined after two-hour dissolution using the CFR method (Table 4). The E% increased with an increase in the hydrophilicity of the formulation (attained by increasing the ratio of P407 and sorbitol, mannitol, or xylitol). Furthermore, there was no significant difference in any of the formulations between the percentage of drug release and E% (*p* > 0.5), which indicated that erosion played a significant role in CHD release. SEM micrographs show the microporous structure of the hydrated tablets (Appendix A); pores would permit water penetration and, consequently, drug release by diffusion [50]. This is confirmed by the swelling of the tablets (Appendix A), which swelled up to 3–4.5 of their origional weight. However, it is not known if the release was mainly controlled by erosion, diffusion, or both. All formulations developed a porous structure when examined under scanning electron microscopy (Appendix A). 

### 2.5. Kinetics of Drug Release

CHD release data were fitted to zero order, first order, Higuchi, Hixon–Crowell, Korsmeyer–Peppas (KP, the power law) and Hopfenberg (HP) models. DDsolver was used to obtain the non-linear fitting coefficients of determination R^2^. MSC was obtained for KP, HP, and zero order due to their comparable and high R^2^ (Table 5). The higher the MSC, the better the fit, and a value greater than 2–3 indicates a good fit [51]. The values of R^2^ for first order, Higuchi, and Hixson–Crowell models are listed in the Appendix A).

In the KP model, the mechanism of drug release is explained by the value of the exponent *n*. This value for the cylindrical tablet is as follows; 0.45> *n* > 0.89 indicating anomalous, 0.89 is case II transport, and >0.89 is supercase II [52]. The latter means that drug release is controlled by swelling and relaxation or erosion [53]. The *n* value for CHD tablets lies between approximately 0.7–1.0.

The exponent *n* in HP model reflects the geometry of the releasing system. The values 1, 1.5, 2, and 3 represent a slab, a half sphere, a cylinder, and a sphere, respectively [54]. DDsolver linearization was performed by starting with *n* = 1 and steadily increasing it to achieve the best linearity [51]. Consequently, a higher *n* value indicated the fit was not valid (Table 5). For instance, the *n* value for S1 obtained using Apparatus 1 was 181.2, and this shape is not valid according to the HP model [54].

Taking into consideration the R2 and MSC (Table 5), the release from S2, S4, M3, M4, X3, and X4 using Apparatus 1 are best fitted with the HP model, which indicates erosion-controlled drug release. The sorbitol formulations with a higher sugar ratio showed erosion control due to the solubility of sorbitol being 2 mg/mL compared to 0.63 and 0.18 mg/mL for xylitol and mannitol, respectively. For the xylitol and mannitol formulations, the erosion was attributed to the high P407 ratio. It is not fully known why the S3 formulation did not follow the same mechanism. The *n* values for erosion controlled formulations was approaching 2 due to the full exposure of the tablet to the dissolution media reflecting its cylindrical shape [54]. The release from S4, M4, and X4 using CFR was mainly controlled by erosion, and it followed zero order drug release and HP model with the *n* value closer to 1. This is because of the release being performed from one surface due to the adherence of the tablet to the sample holder. This might explain the low SI of these formulations, among others (Appendix A), which is attributed to continuous erosion during the swelling process. All other formulations show a better fit with KP model, which indicates the involvement of diffusion, swelling and erosion, or polymer relaxation. The erosion of the tablet is affected by rotation speed, and this explains the higher drug release and why more formulations fit to HP model using Apparatus 1 compared to CFR [55].

### 2.6. Dissolution Efficiency (DE%)

DE was used to compare the two methods of dissolution. DE for CHD release using Apparatus 1 was higher for all formulations, and it was double for S1, S2, M1, M2, X1, and X2 using Apparatus 1 compared to the CFR method. This is attributed to the effect of rotational speed and a higher volume of dissolution media. In contrast, the difference was much less for S4, M4, and X4, with a value of 5%, 3%, and 7%, respectively, which is attributed to its faster hydration and the formation of an interlocked gel layer. (Table 6).

Drug release studies for S4, M4, and X4 showed the best release, and they were least effected by the different methods of drug release.

## 3. Materials and Methods

### 3.1. Materials

The following materials were purchased from Sigma Aldrich, UK: sabauroud dextrose agar (SDA), Muller Hinton broth (MHB), chlorhexidine diacetate salt hydrate (CHD), RPMI-1640 dry powder (RPMI), 3-(*n*-Morpholino) propane sulfonic acid (MOPS), acetone Dulbecco’s Modified Eagle’s Medium (DMEM), a high glucose powder, bovine serum (FBS), L-Glutamine solution (200 mM), Antibiotic-Antimycotic Solution (10,000 U penicillin, 10 mg streptomycin, and 25 μg amphotericin B per ml), Neutral red (3-amino-7-dimethylamino-2-methyl-phenazine hydrochloride Poloxamer 407 (P407), and sorbitol. The following materials were purchased from Alpha Easer, UK: 2,3-bis(2-methoxy-4-nitro-5-sulfophenyl)-2H-tetrazolium-5-carboxanilide (XTT), menadione (2-Methyl-1,4-Naphthoquinone or vitamin K3), and glutaraldehyde and magnesium stearate sorbitol. The following materials were kindly provided by Roquette Company, UK. Xylitol (Xylisorb XTAB 240) and mannitol (Pearlitol 200 SD)., UK: Methocel^TM^ F4M premium hydroxypropyl methylcellulose was kindly provided by Dow-Colorcon Company, UK. 

### 3.2. Methods

#### 3.2.1. Antifungal Assays

##### Minimum Inhibitory Concentration (MIC) and Minimum Biocidal Concentration (MBC)

*C. albicans* ATCC 10,321 were grown on a SDA plate and incubated at 30 °C for 18–24 h. Colonies of *C. albicans* were then transferred to MHB, and the turbidity was adjusted to 0.5 McFarland standard solution. MIC was measured using the broth tube dilution method. CHD was dissolved and serially double-diluted from 10 to 0.087 μg/mL in MHB to a final volume of 5 mL. Then, 200 µL of the inoculum was transferred to each test tube and incubated at 30 °C for 24 h. The MBC was determined by inoculating SDA plates with broth from MIC tubes that showed no visible growth and incubated at 30 °C for 24–48 h. The test was performed in triplicate.

##### *C. albicans* Biofilm Formation and Treatment 

RPMI was prepared as instructed by the manufacturer, and the pH was adjusted to 7.2 using 0.165 M MOPS and seeded with C. albicans. Briefly, a Candida biofilm was formed by adding 200 µL of the suspension containing 5 × 10^5^ CFU/mL to each well of the sterile 96-well microtiter plates and incubated at 37 °C for 4 h (initial biofilm formation) and 24 h (mature biofilm) [56]. Then, the RPMI was removed, and the biofilms were washed twice with 200 µL of phosphate buffer saline (PBS) (10 mM, pH = 7.4) to remove any non-adherent cells. The biofilms were then incubated with CHD at a concentration range of 160 to 0.63 μg/mL for two hours at 37 °C. Finally, the CHD was removed, and the biofilms were washed with 200 µL of PBS.

##### XTT Reduction Assay

The XTT reduction assay measures biofilm viability based on cellular metabolic activity. It has been described in detail elsewhere [56]. Briefly, XTT was dissolved in PBS (pH 7.4) at 0.5 mg/mL, and 10 µL of menadione (10 μM in acetone) was added to 10 mL of XTT solution to facilitate the reduction of the tetrazolium to formazan. Then, 100 µL of XTT/menadione solution was added to each well, and the plate was incubated in the dark for two hours at 37 °C. Subsequently, 80 μL aliquots were pipetted from the supernatant of each well into a new microtiter plate, and the absorbance was recorded at λ_450nm_ using a spectrophotometric plate reader (BioTec, EL800, Swindon, UK). Six replicates were tested.

##### Candida Survival after Treatment 

Viable counting was used to determine the number of living cells, capable of reproduction after treatment with CHD. To each prewashed biofilm, a 100 µL of MHB was added, and the biofilm was resuspended. The suspension was diluted and sub-cultured onto SDA plates, which were then incubated for 24–48 h at 30 °C. The viability was calculated as a percentage count of untreated biofilm. The test was performed in triplicate.

#### 3.2.2. Cytocompatibility

##### Cell Line and Culture Medium

Human embryonic kidney cells (HEK293 cells) were used to test in vitro cytocompatibility. DMEM was supplemented with 10% FBS, 1% L-Glutamine solution (200 mM), and 1% Antibiotic-Antimycotic Solution.

##### Cytocompatibility Assay

HEK293 cells with a density of 4.0 × 10^4^ cells were placed in each well of a 96-well microtiter plate, which was then incubated for 24 h at 37 °C and humidified with 5% CO2. The overnight cell culture medium was replaced with an increasing concentration of CHD in DMEM (1.25–160 µg/mL) and incubated for two hours. The supernatant was aspirated, and cell viability was evaluated using neutral red assay.

NR assay has been described in detail in an earlier study [26]. Briefly, 100 µL of NR (80 mg/L in DMEM) was added to each well and the plate was incubated for two hours at 37 °C. The NR was removed, and the cells were washed in 150 µL of PBS. This was followed by fixation with 100 µL of 5% glutaraldehyde for 2 min. Next, 150 µL of de-stain solution (50% absolute ethanol, 48% ultrapure water, and 2% glacial acetic acid) was added to each well, and the plate was left for 30 min on an orbital shaker to extract the NR from the cells. Finally, the optical density was measured at λ_540n_m (Multiskan Ascent, Thermo Labsystems, Finland). Nine replicates were tested.

#### 3.2.3. Tablet Preparation

CHD buccal tablets were prepared as shown in Table 1. CHD; HPMC; P407; and sorbitol, mannitol, or xylitol powders were blended for 15 min in a V-shaped powder blender (V-1, CapsulCN, China). The dosage of CHD was selected based on the results of antifungal effect and cytocompatibility testing, taking into account safe and effective concentrations.

#### 3.2.4. Powder Flow

##### Flowability (Mass Flow)

The flow rates of the powder blends and granules were measured using an ERWEKA Granulate & Powder Flow Tester (GTL, Regensburg, Germany). Approximately 12 g were placed in the hopper with an orifice of 10 mm in diameter. The results were displayed in g/sec. The test was performed in triplicate.

##### Compressibility Index

The bulk and the tapped volume of the powder blends and the granules were measured using a tapped density tester (TD 50-1000, Varian, UK). The tapped volumes were obtained by subjecting the powders to 200 taps each time until a constant volume was achieved.

Carr’s compressibility index (CI) was calculated using the flowing equation:(1)$$Carr^{\prime }s\;Index=\left(1-V_{t}/V_{0}\right)$$
where *V*_0_ is bulk volume, and *V_t_* is the tapped volume.

Measurements of mass flow and the calculation of the compressibility index were undertaken in triplicate.

##### Melt Granulation

The powder blend was placed in a beaker in a prewarmed water bath at 57 °C, which is equal to the melting point of P407, and mixed with a spatula for 1–3 min, then, sieved through a #18 screen (Appendix A).

##### Tablet Pressing

Magnesium stearate 0.5% was added to the granules and mixed for 3 min prior to being pressed. The granules were then pressed into 6 mm diameter tablets using a ZPS-8 mini rotary tablet machine (Shanghai Tianxiang and Chentai (STC) Pharmaceutical Machinery Co. Ltd., Shanghai, China).

#### 3.2.5. Characterisation of Buccal Tablets

##### Friability

The tablets were characterised for friability using a friability tester (Charles Ischi AG, AE-1, Derbyshire, UK).

##### Tensile Strength

The mechanical strength of the biconvex tablets was calculated using the following equation [57]:(2)$$\sigma_{x}=10\;F/\pi D^{2}{\left[\;2.84H/D-0.126H/W+3.15W/D+0.01\right]}^{-1}$$
where $\sigma_{x}$ is the tensile strength, F is breaking force (Newtons), *D* is tablet diameter, *W* is cylinger length, and *H* is overall thickness of convex-faced disc, with all tablet dimensions in mm.

The tablet breaking force or hardness was measured using a Varian VK 200, (UK) hardness tester (*n* = 10).

##### Swelling Index (SI) 

A randomly selected tablet (*W*_0_) was weighted individually and adhered to a glass slide with a water droplet. The tablet was left for 30 s for the tablet to adhere and was then placed vertically in a beaker in a 37 °C water bath (Clifton, Nickel-Electro Ltd., Bristol, UK). The slide was removed from the water bath and the excess water was dried carefully with a filter paper. The weight (*W_s_*) was recorded at 30, 60, 90, and 120 min, and the SI was calculated using the equation below. Each tablet formulation was assessed in triplicate, and the results are expressed as the mean ± SD.
(3)$$SI=\left(W_{s}-W_{0}\right)/W_{0}$$
where $W_{0}$ is the initial weight and $W_{s}$ is the final weight of the swollen tablet.

##### Determination of Ex Vivo Residence Time

Mucoadhesion was determined as a function of the residence time of the tablets in the oral cavity [11,58]. Chicken crop was used to test the mucoadhesion of the tablets. This is the dilated part of the oesophagus and is lined with a mucous membrane. It was cut into small pieces of approximately 1 × 2 cm, which were then fixed to a glass slide using cyanoacrylate glue. Each tablet was placed on top of the tissue and left for 30 s to start the adhesion process. The glass slide was then placed into the disintegration apparatus Varian VK 100, UK), prefilled with 37 ± 0.2 °C of water [30]. The test was performed for 120 min at 50 rpm, and samples were examined every 15 min. Each tablet formulation was assessed in triplicate.

#### 3.2.6. In Vitro Dissolution of CHD (Apparatus 1) 

The dissolution was performed using Apparatus type 1 (basket method, Varian 705 DS BP Dissolution, Varian, UK). One tablet was placed in 500 mL of water at 37 °C and a rotating speed of 50 rpm. A volume of 4 mL was withdrawn at predetermined time intervals. The concentration of CHD was determined spectrophotometrically at λ_254nm_ (Biochrom WPA Biowave II UV-Spectrophotometry, UK). The test was performed in triplicate.

#### 3.2.7. In Vitro Dissolution Using a Controlled Flow Rate (CFR)

The dissolution was performed under a constant flow rate of 1 mL/min for two hours. The normal salivary flow rate is ≥1 mL/min with a maximum value of 7 mL/min [21,22]. The aim of this investigation was to mimic the salivary flow rate in the oral cavity. Figure 6 illustrates the set-up of the CFR method. Water was placed in a Schott bottle inside a water bath set at 37 °C. The flow rate of the water was controlled using a peristaltic pump at 1 mL/min. A pre-weighed tablet was adhered using one drop of water to a pre-weighed sample holder (the head of a plastic Pasteur pipette, cut into two pieces) and placed in a beaker in the water bath. Samples were collected with a syringe every 10 min for two hours and measured using a UV-VIS spectrophotometer at λ_254_nm. The drug release was recorded for each time point, and the cumulative drug release was calculated by adding the percentage of drug release to the previous time point(s). The test was performed in triplicate.

#### 3.2.8. Tablet Erosion E%

Tablet erosion was investigated to measure the weight of the dried tablet after CRF dissolution. Tablet residues were dried in the oven at 37 °C for 24–48 h, and the percentage erosion (*E%*) was calculated using the following equation.
(4)$$E\%=\left(\;W_{0}-W_{e}\right)/W_{0}\times 100$$
where *W*_0_ is the initial weight of the tablet, and *We* is the dried weight of the tablet after swelling or dissolution.

#### 3.2.9. Dissolution Efficiency (DE%)

DE% is determined from the percentile ratio of the area under the curve (AUC) of the dissolution data to the AUC of 100% release between time zero to the end of dissolution [59]. It was obtained using the equation below. OriginPro 2017 software was used to calculate the area under the curve.
(5)$$D.E.=\frac{\int_{0}^{t}y\times dt}{y_{100}\times t}\times 100\%$$
where *t* is the time, *y* is the percentage of drug dissolved at time *t*, and $\overset{t}{\underset{0}{\int }}y\times dt$ is the area under the curve.

#### 3.2.10. Kinetics of Drug Release

The kinetics of drug release was investigated using zero order (where drug release was at a constant rate), first order (where drug release was concentration-dependant and was proportional to the amount of drug in the matrix), Higuchi (where drug release was difusion-controlled), Hixon–Crowell, Korsmeyer–Peppas models or the power law (where drug release was attributed to diffusion, swelling and erosion, or polymer relaxation), and the Hopfenberg model (where drug release was erosion-controlled) [60]. The latter is an empirical model used to describe drug release from erodible matrices (see the Appendix A). Non-linear fitting for all models was performed using DDsolver, which is an add-in program for Microsoft Excel [51]. To select the best fitting model, both the coefficient of determination R^2^ and the model selection criterion (MSC) were obtained using DDsolver.

#### 3.2.11. Fourier Transform Infrared Spectroscopy (FTIR) and Differential Scanning Calorimetry (DSC) 

FTIR and DSC were used to investigate the possibility of interaction between the polymers and the drug. CHD, HPMC, P407, S4, M4, and X4 physical mix, granules, and tablets were investigated. FTIR Spectra were obtained using a Bruker Alpha spectrometer (Germany). Samples were scanned from 4000 to 400 cm^−1^. For DSC, a sample size of 5.0 ± 0.2 mg was heated from 25 to 300 °C in an aluminum pan under a nitrogen flow of 40 mL/min at a scan rate of 10 °C/min. The analysis was performed using Mettler Toledo DSC823e (Switzerland).

#### 3.2.12. Statistical Analysis

The statistical differences were determined using the student’s unpaired t-test (Welch’s), the paired t-test using Grappad Prism 9.1.0. *p* < 0.05 was considered to be statistically significant.

## 4. Conclusions

The exposure of the human cell line HEK293 to a concentration of 20 µg/mL of chlorhexidine for two hours showed no cytotoxic effect and a promising antifungal activity against *C. albicans* biofilms with complete killing of planktonic cells. This concentration was utilized to prepare mucoadhesive buccal tables for targeted buccal drug delivery. CHD is not absorbed from the GIT, and consequently, it has no systemic side effects or drug–drug interaction. Melt granulation showed a remarkable improvement in powder flow, and no interaction was observed between CHD and the excipients. The CFR method was used to mimic drug release in the oral cavity from saliva. The formulations S4, M4, and X4 showed zero order and erosion-based drug release. They can be considered suitable candidates for the treatment of OPC. The formulations can be further investigated for different drugs, trans-buccal, and oral systemic sustained release formulations.

## Figures and Tables

**Figure 1 pharmaceuticals-14-00493-f001:**
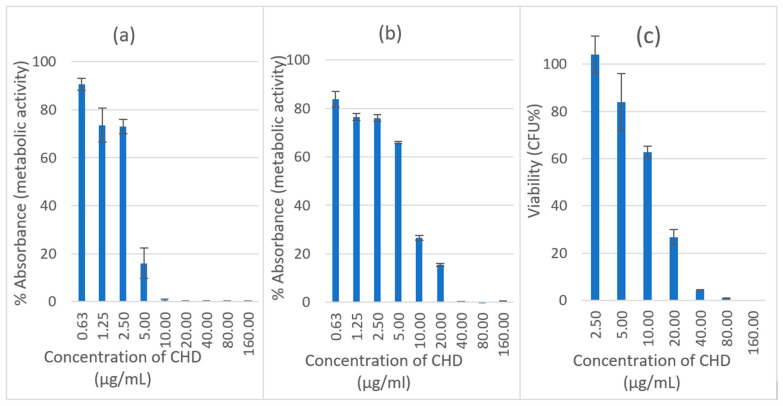
Effect of CHD on initial *C. albicans* biofilm, (**a**) 4 h and (**b**) 24 h using XTT assay. Data are expressed as mean percentages ± SE, *n* = 12. (**c**) Biofilm recovery (viable count) of *C. albicans* after treatment with CHD at 30 °C, Data are expressed as mean percentages ± SE, *n* = 3.

**Figure 2 pharmaceuticals-14-00493-f002:**
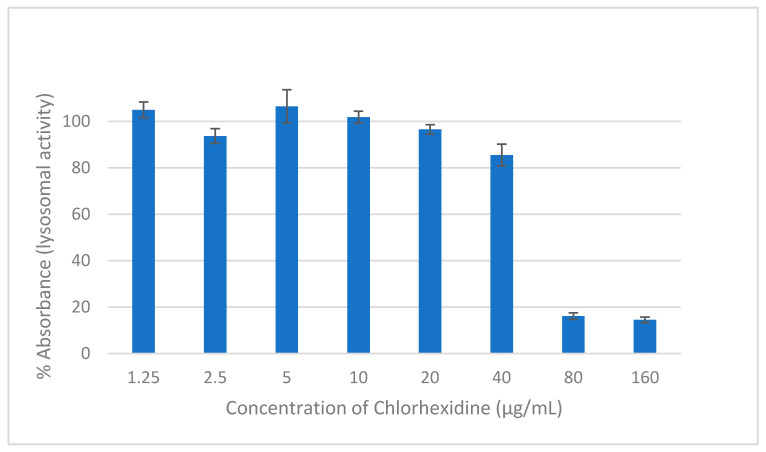
The effect on the viability of HEK293 cells of two hours of exposure to different concentrations of CHD measured using NR assay. Data are expressed as mean percentages ± SE, *n* = 9.

**Figure 3 pharmaceuticals-14-00493-f003:**
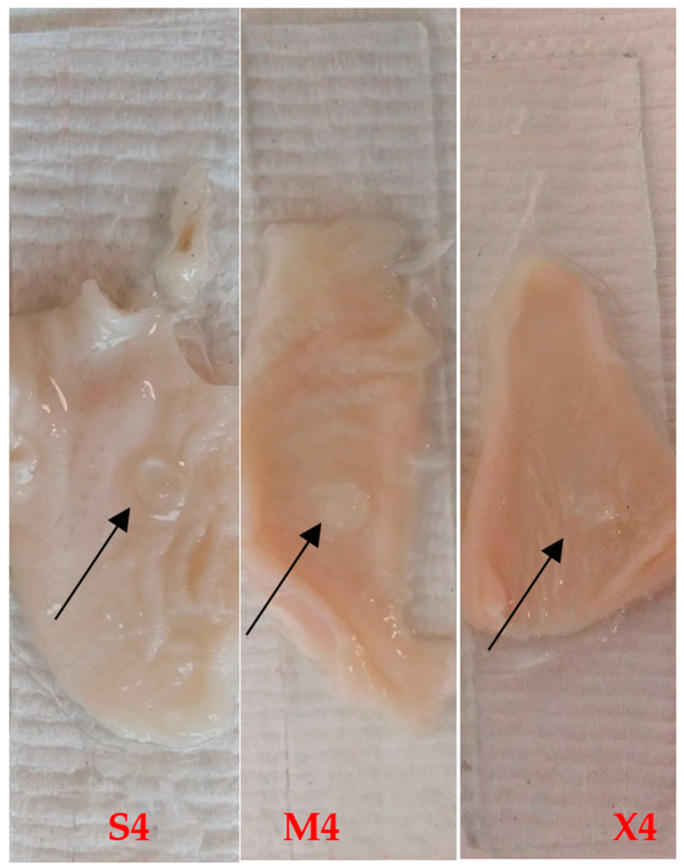
S4, M4, and X4 after two-hour ex vivo mucoadhesion using disintegration apparatus. The arrows show the hydrogel tablet after two hours.

**Figure 4 pharmaceuticals-14-00493-f004:**
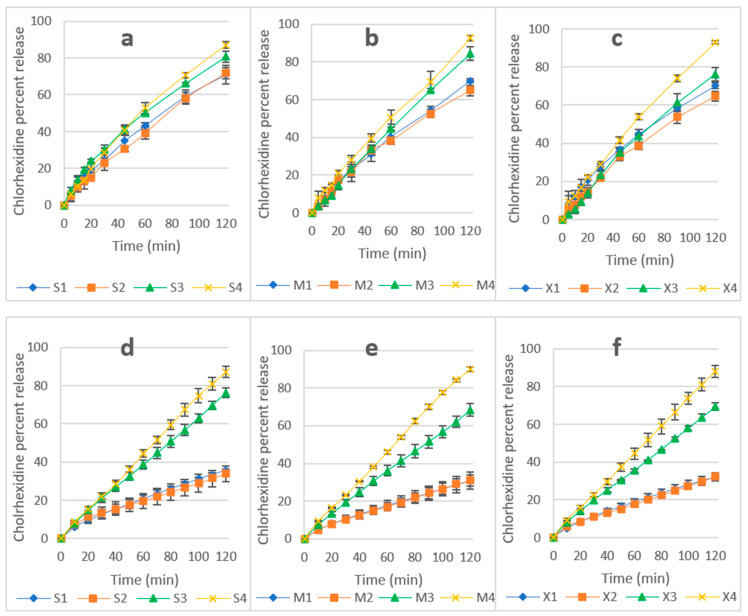
(**a**–**c**) CHD release using Apparatus I (**a**—sorbitol, **b**—mannitol, and **c**—xylitol formulations). (**d**–**f**) CHD release using CFR (**d**—sorbitol, **e**—mannitol, and **f**—xylitol). Data are expressed as mean percentages ± SD, *n* = 3.

**Figure 5 pharmaceuticals-14-00493-f005:**
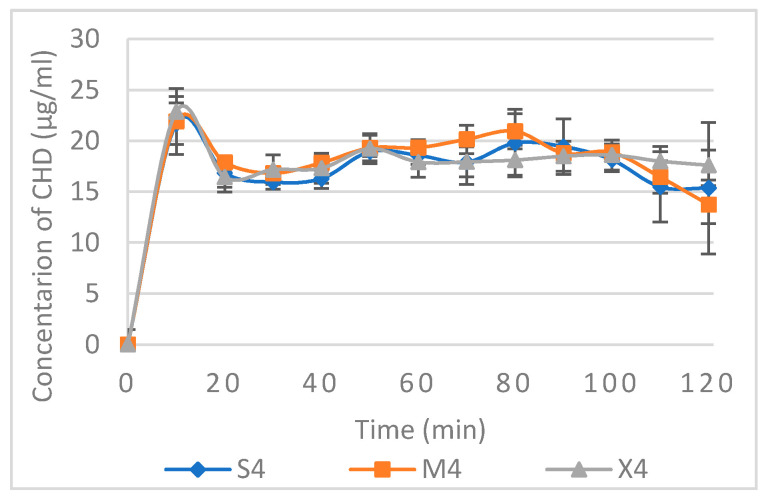
Non-cumulative CHD release from S4, M4, and X4 using CFR 1mL/min. Data are expressed as mean concentration ± SD, *n* = 3.

**Figure 6 pharmaceuticals-14-00493-f006:**
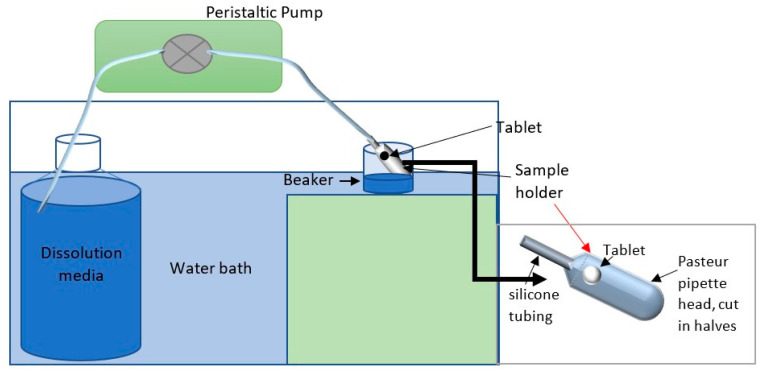
Diagram illustrating the setup of the CFR method.

**Table 1 pharmaceuticals-14-00493-t001:** Composition of CHD mucoadhesive buccal tablets.

Ingredients	S1	S2	S3	S4	M1	M2	M3	M4	X1	X2	X3	X4
	Quantity mg/tab											
CHD	2.5	2.5	2.5	2.5	2.5	2.5	2.5	2.5	2.5	2.5	2.5	2.5
P407	14	14	21	21	14	14	21	21	14	14	21	21
HPMC	14	14	7	7	14	14	7	7	14	14	7	7
Sorbitol	7	14	7	14	-	-	-	-	-	-	-	-
Mannitol	-	-	-	-	7	14	7	14	-	-	-	-
Xylitol	-	-	-	-	-	-	-	-	7	14	7	14
Weight	37.5	44.5	37.5	44.5	37.5	44.5	37.5	44.5	37.5	44.5	37.5	44.5
Ratio (code) P407/HPMC/Polyol	2:2:1	2:2:2	3:1:1	3:1:2	2:2:1	2:2:2	3:1:1	3:1:2	2:2:1	2:2:2	3:1:1	3:1:2

**Table 2 pharmaceuticals-14-00493-t002:** Flow rate (g/sec) of granules through a 10 mm orifice size. Data are expressed as mean ± SD, *n* = 3.

Formulation	Powder Blends		Granules		
	CI				Flow g/s
M1	30.81 ± 0.87	*p*	20.72 ± 0.61	F	4.73 ± 0.12
M2	37.20 ± 0.073	VP	20.57 ± 2.05	F	4.73 ± 0.12
M3	40.07 ± 0.66	VVP	12.50 ± 0.00	G	6.30 ± 0.00
M4	29.60 ± 0.37	*p*	11.81 ± 0.16	G	6.30 ± 0.00
X1	39.11 ± 1.32	VVP	20.86 ± 1.04	F	4.43 ± 0.15
X2	34.97 ± 4.23	VP	22.72 ± 1.09	PA	4.53 ± 0.12
X3	35.43 ± 3.81	VP	21.79 ± 1.43	PA	4.67 ± 0.23
X4	39.47 ± 2.70	VVP	21.81 ± 1.42	PA	6.00 ± 0.30
S1	39.50 ± 2.09	VVP	22.76 ± 1.05	PA	4.53 ± 0.15
S2	39.87 ± 1.66	VVP	22.32 ± 1.74	PA	4.80 ± 0.20
S3	39.32 ± 1.65	VVP	27.46 ± 1.37	*p*	6.83 ± 0.23
S4	41.55 ± 3.57	VVP	22.88 ± 1.26	PA	6.10 ± 0.00

G good, F fair, PA passable, P poor, VP very poor, VVP very very poor.

**Table 3 pharmaceuticals-14-00493-t003:** Friability and tensile strength of CHD mucoadhesive buccal tablets.

	Tensile Strength (MPa) *n* = 10 ± SD	Friability (%)
S1	1.03 ±0.29	0.00
S2	0.80 ± 0.23	0.07
S3	0.90 ± 0.28	0.17
S4	0.85 ± 0.14	0.01
M1	0.63 ± 0.35	0.19
M2	0.94 ± 0.16	0.08
M3	0.96 ± 0.22	0.33
M4	0.88 ± 0.18	0.26
X1	0.37 ± 0.06	0.26
X2	0.67 ± 0.11	0.10
X3	0.56 ± 0.17	0.23
X4	0.74 ± 0.21	0.08

**Table 4 pharmaceuticals-14-00493-t004:** E% and CHD% release using the CFR method at the two-hour point (± SD, *n* = 3).

Formulations	% E (2 h)	% CHD Release (2 h)	Paired *t* Test
S1	36.87 ± 5.88	35.57 ± 0.76	No significant difference *p* > 0.5
S2	39.70 ± 9.04	33.87 ± 4.16	
S3	70.84 ± 10.17	76.14 ± 2.52	
S4	88.15 ± 2.62	87.18 ± 2.99	
M1	28.43 ± 3.12	30.97 ± 4.58	
M2	28.86 ± 1.69	30.92 ± 2.92	
M3	69.50 ± 5.65	68.41 ± 3.40	
M4	84.63 ± 4.07	89.81 ± 1.22	
X1	30.41 ± 5.25	32.14 ± 2.35	
X2	36.44 ± 2.54	32.31 ± 1.00	
X3	70.41 ± 1.67	69.35 ± 2.03	
X4	87.21 ± 5.67	88.05 ± 3.26	

**Table 5 pharmaceuticals-14-00493-t005:** Drug release kinetics using different fitting models for CHD formulation (Apparatus 1 dissolution) and CFR 1mL/min.

		ZERO		KP			HP		
		*R* ^2^	MSC	R^2^	MSC	*n*	*R* ^2^	MSC	*n*
Apparatus 1	S1	0.978	3.3	0.994	4.6	0.750	0.992		181.8
	S2	0.992	4.3	0.995	4.2	0.848	0.997	4.6	2.2
	S3	0.979	3.4	0.998	4.6	0.708	0.991		248.6
	S4	0.985	3.6	0.995	4.7	0.822	0.996	4.9	2
	M1	0.988	4.1	0.995	4.4	0.856	0.996		6.8
	M2	0.975	3.2	0.985	3.4	0.819	0.991		291.1
	M3	0.996	5.1	0.992	4.4	1.047	0.997	5.5	1.1
	M4	0.993	4.2	0.994	4.2	0.825	0.993	4.4	1.4
	X1	0.970	2.9	0.997	5.0	0.694	0.983		1344.8
	X2	0.980	3.4	0.991	3.9	0.826	0.995		99.5
	X3	0.988	3.9	0.991	3.9	1.058	0.996	5.1	1.7
	X4	0.992	4.3	0.995	4.5	0.815	0.995	4.8	1.6
CFR 1 mL/min	S1	0.984	3.9	0.998	5.7	0.756	0.980		425.7
	S2	0.975	3.1	0.993	4.3	0.677	0.951		605.6
	S3	0.998	5.9	0.999	6.9	0.941	0.998	6.0	1.2
	S4	0.998	5.9	0.998	5.4	0.992	0.998	5.8	1.0
	M1	0.991	4.2	0.999	6.9	0.783	0.990		671.8
	M2	0.992	4.6	0.998	5.6	0.813	0.989		177.3
	M3	0.998	6.0	1.000	7.7	0.895	0.999	6.2	1.4
	M4	0.999	6.4	0.999	5.6	0.962	0.999	6.5	1.1
	X1	0.988	3.9	0.999	6.7	0.766	0.988		495.6
	X2	0.992	4.2	0.996	4.7	0.792	0.986		438.6
	X3	0.998	6.0	0.999	6.6	0.901	0.998	6.0	1.3
	X4	0.999	6.6	0.999	6.4	0.956	0.999	6.5	1.0

**Table 6 pharmaceuticals-14-00493-t006:** DE of CHD release using Apparatus I and CFR methods. Data are expressed as mean percentages ± SD, *n* = 3.

Formulations	Apparatus I	CFR
S1	41.12 ± 2.46	20.25 ± 1.48
S2	39.12 ± 2.72	19.54 ± 3.46
S3	47.34 ± 0.73	38.78 ± 2.06
S4	48.89 ± 1.11	44.44 ± 1.92
M1	38.02 ± 2.49	17.19 ± 2.71
M2	36.85 ± 1.30	16.94 ± 2.39
M3	43.48 ± 0.63	35.31 ± 2.39
M4	49.03 ± 2.82	46.34 ± 0.54
X1	42.10 ± 0.46	18.16 ± 1.57
X2	37.19 ± 1.68	17.57 ± 0.43
X3	41.52 ± 2.79	35.74 ± 0.68
X4	51.40 ± 0.46	44.55 ± 2.34

## Data Availability

Not Applicable.

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
