# Peer review of "Chlorhexidine Mucoadhesive Buccal Tablets: The Impact of Formulation Design on Drug Delivery and Release Kinetics Using Conventional and Novel Dissolution Methods"

_pharmaceuticals, 2021, doi:10.3390/ph14060493_

Round 1

Reviewer 1 Report

Authors have carefully addressed all the suggestions and comments.

Author Response

Dear Reviewer,

Thank you for your time and consideration.

Regards

Reviewer 2 Report

In the present manuscript mucoadhesive hydrogel buccal  tablets containing HPMC and P407 polymers were prepared to control the release of CHD. The results showed that the prepared formulations can be considered as suitable candidates for the treatment of OPC. To my opinion the manuscript should be published in Pharmaceuticals after  minor revision.

Minor comments:

1.Introduction: Regarding the mucoadhesive hydrogel materials for the treatment of OPC, a more detailed discussion and appropriate referencing is required. The authors have to present other classes of mucoadhesive carriers that have been studied such as cellulose derivatives, chitosan etc.

2.Provide a reference for the setup of the CFR method

Reviewer 3 Report

This paper deals with the evaluation of Chlorhexidine and a prepared mucoadhesive buccal tablet as fungicide against Candida biofilms. Tablets formulation was optimized by analyzing the physicochemical properties, mucoadhesion and drug release kinetics. The investigation is properly addressed, the manuscript is well written and data are clearly presented. However, there are some points that must be improved:

Information related with employed polymers and the formation of the hydrogel seems to be not correct. In these formulations, the polymer that promotes hydrogel formation is the poloxamer, due the thermogelation at room temperature (as authors mention). However, this point is not clear in the manuscript. In addition, incorrectly, authors state in line116 that “hydrogels polymers..” were used, and in line 127, that HPMC has hydrogel forming properties, which is misleading, because authors do not have the corroboration that HPMC is crosslinked with poloxamer. In the bibliography, HPMC is described in this formulation as excipient (example: In situ gel forming systems of poloxamer 407 and hydroxypropyl cellulose or hydroxypropyl methyl cellulose mixtures for controlled delivery of vancomycin. Journal of Applied Polymer Science 109(4):2369 – 2374).

-Lines143-145 must been rewritten for clarifying.

-Results related to table 3 are described (152-157) but they are not discussed and explained according to the specific physicochemical properties of the formulations.

-Lines 159-161. Swelling differences are explained in terms of viscosity, instead of hydrophilicity (content of HPMC). Besides, the decrease of swelling with polyols ratio controversely is explained by the higher solubility, which must be the opposite. Thus, this section must be clarified.

-Interaction between negatively charged HPMC and the drug has not been considered along the discussion on release profiles.

Round 2

Reviewer 3 Report

Proposed changes have been properly addressed. I did not understand some points that have been now clarified by authors.

This manuscript is a resubmission of an earlier submission. The following is a list of the peer review reports and author responses from that submission.

Round 1

Reviewer 1 Report

The present paper reports on the study and the development of chlorhexidine mucoadhesive buccal tablets prepared by compaction of particles prepared by melt granulation. The aim of the work is interesting but the work is not soundly performed. Therefore, in my opinion, it should be rejected:

- The authors should explain how to verify the safety of the prepared dosage forms bi in vitro and ex vivo experiments

- The authors should justify the choice of formulation components and their amount

- The authors should explain why they chosen the melt granulation process. Did they consider the stability issues due to the temperature applied?

- The authors could analyse the results obtained by applying the methods linked to Design of Experiments methodology

- The swelling profiles could be more useful with respect to other results shown

- Paired t test results should be better explained

- The authors should describe the choice of different kinetic models applied

- some of the methods should be described more in depth

Reviewer 2 Report

The manuscript by Enas Al-Ani et al., reports about the preparation and characterization of several buccal tablets by varying the ratios of hydroxypropyl methylcellulose, poloxamer 407 and 3 different poliols. Tablets are loaded with chlorhexidine for the treatment of oropharyngeal candidiasis as drug efficacy and safety at the selected concentration were assessed. Even the proposed paper is interesting and the general concept is well designed some experiments needs to be clarified as well as the discussion section.

Some crucial points:

1) ABSTRACT:

- line 15: I guess that chlorhexidine activity against Candida albicans biofilms and cytotoxicity against the HEK293 human cell line was evaluated. However the sentence is confusing as it seems to refer to the buccal tablet while their preparation has not been discussed yet.

- as authors are not proposing an anticancer formulation it should be better mention cytocompatibility instead of cytotoxicity.

2) INTRODUCTION:

- authors refers to buccal tablet as smooth and soft. Are you sure that you are not talking about buccal films?

- lines 45-64 and 68-69: the discussion that explain the reasons why all the various components of the formulation were chosen should be moved in the results and discussion section.

3) RESULTS AND DISCUSSION:

- in this section authors refer to table 6 immediately while table 6 is pages away. Please move table 6 in order to avoid confusion.

- I am not sure about the methodology for the mucoadhesion evaluation. There are no references and the obtained data are not described. Please clarify both the assay and the obtained results.

- line 162: actually the controlled flow method is not new: some examples https://doi.org/10.1016/j.ejpb.2007.02.020, https://doi.org/10.3109/03639045.2014.971030, DOI: 10.3390/jpm10040242.

- figure 4: numbers of the y axis and text are superimposed.

- lines 169-187: the discussion regarding dissolusion data (figure 4) is confusing. Authors claim that “Sorbitol, mannitol and xylitol did not improve the release from these formulations” however no studies were conducted without poliols and thus the only possible conclusion is that by varing the poliols the drug release behavior still remain the same. Moreoever line 172 (Sorbitol, mannitol and xylitol did not improve the release from these formulations) is opposite to line 183 (Sorbitol, mannitol and xylitol improved CHD release due to their high solubility). In addition authors claim that the two methods to evaluate drug dissolution give similar results, however observing the proposed graphs it does not seem so (a ≠ d, b ≠ e, c ≠ f) as in terms of percentage of drug released from the various formulations as in terms of shape of the release curve. Please try to correct and complete the discussion.

- How were E% and DE% calculated?

- I suggest to move all the fitting data in the main text.

- It is not clear the reasons why the two methods to evaluate drug dissolution were employed and consequently how to argue the results obtained and the differences observed between the two methods.

4) MATERIALS AND METHODS:

- refer to cytocompatibility instead of cytotoxicity.

- please provide two separate paragraph: materials, methods.

- please provide the number of each experiment repetitions.

5) SUPPLEMENTARY MATERIALS:

- the SEM analysis appear only in this section and are never mentioned in the main text.

Reviewer 3 Report

The authors have designed and characterized several Chlorhexidine mucoadhesive buccal tablets.  The research objectives are good however there are number of lacunae and queries, which need to be addressed:

  • Abstract:

it is really confusing and should better summarize all the results. It needs to be completely rewritten with greater scientific rigor.

  • Introduction

 should be improved because it does not provide sufficient background.  In fact the part concerning the reasons for the choice of materials to formulate the tablets described in the work should be moved to “results and discussion” section.

  • Results and discussion

Line 77: The acronyms should be explained

  • 4. Characterisation of powder blends and granules

Line 126: Table 6  must become table 1 and be inserted at the end of the sentence.

  • 4.3. Ex-vivo mucoadhesion

The evaluation of mucoadhesion is very questionable. Not parameter, not numeric evaluation and not references are reported for the method adopted by authors. Why for Authors the mucoadesion obtained is “ideal”? what parameter did they measure? This paragraph should be deleted or enriched with scientific data.

  • 4.4. In-vitro dissolution and erosion studies

Line 162 The acronyms should be explained

Line 162 “a new method” is not new! (e.g. doi:10.3390/pharmaceutics9030022)

Graphics of figure 3 are confusing in y-axis. Also legend should be changed and better explained.

Figure 3 is not mentioned along the test. this leads to not understanding to which figure the sentence in line 169 refers

Line 169: Which figure 4?

From line 170 to line 189 the discussion is very very confusing. Authors must re-phrase the entire paragraph to make their results understood.

Why Authors use two different methods for evaluating tablet dissolution? The reasons should be explained.

  • Materials and Methods  

Paragraph “Materials” was omitted

Number of replicate are omitted

Statistical analysis was omitted

Line 333: brand and model of V-shaped powder blender was omitted

There are typos in table 6

Line 344: model of density tester was omitted

  • 4.4. Tablet pressing

Model of machine and the compressive force used were omitted.

  • 4.5.2. Tensile strength

The parameters described in the equation do not correspond to those reported in the reference. F for the authors is the hardness, in the reference it is the load; W is the central cylinder tickness while in the note is the length of the cylinder. Authors should clarify.

  • 4.5.3. Swelling Index (SI)

Authors should improve the description of test.  Some details were omitted.

      4.5.4. Ex-vivo mucoadhesion analysis  

The comments are reported above!

SUPPLEMENTARY MATERIALS:

- the SEM analysis was reported in this section and is not mentioned in the main text.